# Elucidating Transmission Patterns of Endemic *Mycobacterium avium* subsp. *paratuberculosis* Using Molecular Epidemiology

**DOI:** 10.3390/vetsci6010032

**Published:** 2019-03-20

**Authors:** Rebecca M. Mitchell, Annabelle Beaver, Elena Knupfer, Abani K. Pradhan, Terry Fyock, Robert H. Whitlock, Ynte H. Schukken

**Affiliations:** 1Emory University Department of Mathematics and Computer Science, 400 Dowman Drive, Atlanta, GA 30322, USA; rmm257@gmail.com; 2Department of Population Medicine and Diagnostic Sciences, Cornell University, 618 Tower Road, Ithaca, NY 14853, USA; ab2368@cornell.edu (A.B.); eknupfer85@gmail.com (E.K.); 3Department of Animal Science, Cornell University, 507 Tower Road, Ithaca, NY 14853, USA; 4Quantitative Veterinary Epidemiology Group, Wageningen University and Research, Wageningen, Radix Building, Droevendaalsesteeg 1, 6708PB Wageningen, The Netherlands; 5Department of Nutrition and Food Science, and Center for Food Safety and Security Systems, University of Maryland, College Park, MD 20742, USA; akp@umd.edu; 6Department of Clinical Studies, University of Pennsylvania, School of Veterinary Medicine, New Bolton Center, Kennett Square, PA 19348, USA; tfyock@vet.upenn.edu (T.F.); rhw@vet.upenn.edu (R.H.W.); 7College of Veterinary Medicine, Utrecht University, 3584 CL Utrecht, The Netherlands; 8GD Animal Health, 7400 AA Deventer, The Netherlands

**Keywords:** *Mycobacterium avium* subsp. *paratuberculosis* (MAP), mycobacterial co-infections, MLSSR typing, mutation rate, within-host evolution, vertical transmission

## Abstract

Mycobacterial diseases are persistent and characterized by lengthy latent periods. Thus, epidemiological models require careful delineation of transmission routes. Understanding transmission routes will improve the quality and success of control programs. We aimed to study the infection dynamics of *Mycobacterium avium* subsp. *paratuberculosis* (MAP), the causal agent of ruminant Johne’s disease, and to distinguish within-host mutation from individual transmission events in a longitudinally MAP-defined dairy herd in upstate New York. To this end, semi-annual fecal samples were obtained from a single dairy herd over the course of seven years, in addition to tissue samples from a selection of culled animals. All samples were cultured for MAP, and multi-locus short-sequence repeat (MLSSR) typing was used to determine MAP SSR types. We concluded from these precise MAP infection data that, when the tissue burden remains low, the majority of MAP infections are not detectable by routine fecal culture but will be identified when tissue culture is performed after slaughter. Additionally, we determined that in this herd vertical infection played only a minor role in MAP transmission. By means of extensive and precise longitudinal data from a single dairy herd, we have come to new insights regarding MAP co-infections and within-host evolution.

## 1. Introduction

*Mycobacterium avium* subsp. *paratuberculosis* (MAP) is the etiologic agent of Johne’s disease, a chronic granulomatous enteritis causing diarrhea and progressive weight loss in ruminant species. Johne’s disease results in an estimated 250-million-dollar loss annually for the U.S. dairy industry, owing to factors such as reduced milk production and premature culling [1]. In addition to adverse impacts on animal welfare and agro-economics, MAP may be of public health significance since it has been implicated as a potential cause of human Crohn’s Disease [2].

MAP is of further relevance from a public health standpoint due to notable parallels between the mechanisms of MAP and *M. tuberculosis* infections. Both types of infection are characterized by lengthy incubation periods resulting in a large proportion of subclinical cases. Additionally, *M. tuberculosis* and MAP are marked by multiple disease presentations [3] and share the macrophage as a primary target for colonization [4]. Finally, the presence of co-infections and their potential clinical importance in both MAP and *M. tuberculosis* suggests that within-host infection dynamics of MAP could have broad application in understanding the pathobiology of multiple mycobacterial infections [4]. 

Calves are most susceptible to infection with MAP [5]. Yet, it is difficult to differentiate vertical transmission (either in utero or immediately post-calving) from two separate infection events in susceptible dam and daughter. In the absence of strain-typing data, this potential genetic susceptibility to MAP of both dam and daughter may be misconstrued as vertical transmission. Heritability of MAP susceptibility has been estimated as low to moderate [6,7]; however as argued here, these estimates are highly dependent upon diagnostic methods, breed, and statistical model selection. Management recommendations for reducing within-herd MAP infections typically focus on culling high-shedding animals and minimizing exposure of newborn calves to potential sources of MAP [8]. These control measures do not account for the differences in transmission dynamics based on horizontal, potentially genetically controlled, compared to vertical transmission. Although these two concepts cannot be completely disentangled, control program recommendations could be tailored according to their relative importance in MAP transmission. For example, culling based upon kinship alone would not be indicated unless genetic susceptibility contributed substantially to overall infection dynamics. Hence, knowledge of the relative impact of transmission routes is of great clinical relevance.

Current multi-locus short-sequence-repeat-based molecular typing methods (MLSSR) allow for distinction between bovine-specific MAP strains within a single farm [9]. Neonatal calves are assumed to be the primary targets of new MAP infection, yet the latency between infection and detection based on current diagnostic testing is typically between 2 and 5 years [10]. Given this latency, determining an estimate for the rate of strain mutation within the host is important to distinguish vertical versus horizontal transmission. Within-host evolution of strains would show in our data as animals infected with multiple SSR types, despite actual vertical transmission. Conversely, animals with multiple SSR types may have been infected from multiple distinct transmission events. Using MAP SSR type diversity in the samples taken from one animal as a proxy for within-host mutation, the evaluation of within-host mutation relative to between-host SSR type diversity may allow potential distinction between in vivo within-host evolution and multiple independent transmission events [11]. Previous work [12,13] has identified cattle infected with multiple MAP MLSSR types, but these data were cross-sectional and lacked a more complete MAP strain distribution in the population of interest; it was consequently impossible to distinguish multiple transmission events from strain evolution over time. Longitudinal data with more or less complete coverage of the MAP infection status of population of interest would be necessary to better understand MAP infection dynamics. 

In this study we therefore use longitudinal MAP infection data in a precisely-documented dairy herd over a period of at least seven years. These repeated-measures data provide an opportunity to study MAP transmission routes with the specific objectives to (1) distinguish between horizontal and vertical MAP transmission and (2) to differentiate multiple new infection events from within-host evolution. 

## 2. Materials and Methods

### 2.1. Study Design and Sample Collection

All animal sampling procedures were approved by the Cornell University Institutional Animal Care and Use Committee. Samples were collected from a commercial dairy farm in New York State as part of a longitudinal study by the Regional Dairy Quality Milk Alliance (RDQMA) [14]. Complete details of the study design including methods of sample collection and processing have been published elsewhere in detail [14,15]. Briefly, the herd was visited semi-annually, and herd records, including individual animal demographics, production data, movements within herd pens, and health information were collected at each visit between January 2004 and December 2011. Individual animal blood and fecal samples were collected semi-annually. The fecal samples were obtained from all animals in first or greater lactation using individual rectal sleeves. The samples were transferred to 50 mL conical tubes and shipped on ice overnight to the University of Pennsylvania for further processing. In addition to the semi-annual whole-herd fecal samples, subsets of additional fecal samples were taken based on prior positive culture results or from otherwise MAP-suspect animals. These samples were shipped and processed in the same manner as the full-herd fecal samples. 

Culled animals were tagged for study identification prior to sale. Study animals were therefore more easily recognized by slaughter house staff. During carcass processing in a large regional processing facility, four types of intestinal tissue samples (lymph nodes (2 samples), ileum, and ileocecal valve) and a fecal sample were collected from the tagged animals by trained staff and placed in individual tubes. These samples were shipped overnight on ice to the University of Pennsylvania paratuberculosis laboratory. 

In summary, the actual sample collection was carried out in a herd of approximately 330 dairy cows over a 7+ year period with sampling of animals every half year. Initial animal demographic, health and production data and follow-up data from animals in the study were available for a period of two years before the first sample was collected and approximately 10 years after the start of sample collection in January 2004. Throughout the study, all serum and culture results were shared with the owner of the farm.

### 2.2. Sample Processing

Fecal samples were decontaminated then cultured for MAP on Herrold’s Egg Yolk Media (HEYM) with Mycobactin J (Allied Monitor, Fayette, MO, USA) using the Whitlock purification and culture protocol [16]. Tissue samples from culled animals were processed using a separate protocol as described in previous work [17,18]. 

MAP shedding was quantified based upon colony growth on 4 HEYM slants. Because high colony density is associated with decreased precision of colony counts, samples with high counts (>75) were assigned CFU ranges based on colony count estimates rather than precise individual counts. Maximum possible per-tube classification was ≥ 300 CFU. Individual colonies from all samples that grew on the original Mycobactin J supplemented HEYM were substreaked onto media with and without Mycobactin. The growth on the Mycobactin tube was usually too numerous to count (TNTC) or confluent, and colonies from that tube were preserved on porous beads, which are intended as carriers to support the viability of microorganisms during storage in the freezer. A single frozen bead was then used to re-inoculate a HEYM tube, representing a third generation of culture. Colonies that were successfully sub-streaked from the original isolation process were submitted for SSR typing. Sub-streaked colonies that grew in the HEYM tube often had many colonies per tube, these multiple colonies were used for SSR typing.

### 2.3. DNA Extraction

DNA was extracted from the bacterial lawn on sub-streaked HEYM slants recovered from a single bacteria-coated bead. A sterile plastic 10 μL loop was used to re-suspend a portion of the bacterial lawn in sterile water (Invitrogen Corporation, Carlsbad, CA, USA), 650 buffer AL (Qiagen, Valencia, CA, USA) and 250 μL of sterile 0.1 mm zirconia/silica beads (BioSpec Products, Inc. Bartlesville, OK, USA) in a 2 mL beadbeater screwcap vial (Biospec Products, Inc., Bartlesville, OK, USA). Samples were processed in small batches of 4–10 with a concurrent negative extraction control as described by Pradhan et al. [15]. Cells underwent disruption for 5 min at maximum speed in a Mini Beadbeater-8 (Biospec Products, Bartlesville, OK, USA). DNA was then extracted and purified using a QIAamp DNA Mini Kit (Qiagen, Inc., Valencia, CA, USA) according to the protocol provided by the manufacturer with the following modifications: after beadbeating, the sample was incubated for 30 min at 70 °C prior to removing 600 μL to combine with 600 μL AL buffer and 60 μL of proteinase K (Qiagen, Valencia, CA, USA). This aliquot was vortexed and incubated for an additional 30 min at 70 °C. 600 μL of 100% ethanol was added to the lysate and vortexed. All lysate was passed through QIAamp Mini columns in 600 μL volumes, which were spun at 6000 g for 1 min. The remainder of the manufacturer protocol was followed until elution. Elution was conducted in 2 steps: 10 min room-temperature incubations with 100 μL and 50 μL of sterile water, respectively, followed by 3 min spins at 6000 g. Eluted samples were stored at −20 °C until PCR typing. 

### 2.4. Short Sequence Repeat Typing (SSR)

SSR typing based on an 11-locus typing scheme [19] was used to determine strain designation. Initially, the four loci (Locus 1, Locus 2, Locus 8, and Locus 9) described by Harris et al. [12] and Pradhan et al. [15] were selected because of high discriminatory power and comparability across studies. Locus 2 was not included in this analysis due to poor sequence resolution with long G repeats. Five additional loci described by Amonsin et al. [19] were evaluated, to increase discriminatory power (locus 3, locus 5, locus 6, locus 10, and locus 11). The discriminatory value of these loci was assessed after processing samples from dam–daughter pairs and a subset of all available samples with a minimum of 50 samples per locus. When diversity was not observed in this subset, the locus was not processed for the remainder of the samples. This was the case with three loci (incl. the number of samples processed per locus): locus 3 (*n* = 58), locus 5 (*n* = 55), and locus 11 (*n* = 61). 

PCRs were carried out with the extracted DNA for all samples using the primers described by Amonsin et al. [19] with one modification. As in Pradhan et al. [15], the following alternate primers for locus 1 were used: 5′-GTG TTC GGC AAA GTC GTT GT-3′ and 5′-GCG GTA CAC CTG CAA G-3′. The 25 μL PCR amplification reaction mixture for each SSR contained: 12.5 μL of 2× GoTaq Green Master Mix (Promega Corporation, Madison, WI, USA), 0.625 μL of 10 μM upstream and downstream primers (Integrated DNA Technologies, Coralville, IA, USA), 9 μL of distilled water, 1.25 μL of DMSO (Dimethylsulfoxide) and 1 μL of template sample. As in our previous work [15], amplification was performed using the following conditions: an initial denaturation at 94 °C for 2 min, followed by 40 cycles of denaturation at 94 °C for 30 s, annealing at 60 °C for 1 min, and extension at 72 °C for 1 min, with a final elongation step at 72 °C for 7 min. The modified locus 1 amplification was performed under the same conditions, but for 35 cycles instead of 40. Negative PCR and extraction controls were included with each run. 

Two μL of each PCR product were electrophoresed at 105 V for 30 min in 0.5× TBE buffer (0.45 M Tris-Borate, 0.01 M EDTA, pH 8.3) on a 1.5% (wt/vol) agarose gel with 5 μL ethidium bromide. PCR products were visualized via UV transillumination on an InGenius Gel Documentation System (SynGene, Frederick, MD, USA). Amplicons were purified using either a PureLink PCR purification kit (Invitrogen, Carlsbad, CA, USA) or QIAquick PCR purification kit (Qiagen, Valencia, CA, USA) and quantified using a NanoDrop ND-1000 spectrophotometer (NanoDrop Technologies Inc., Wilmington, DE, USA). PCR amplicons were sequenced using standard dye terminator chemistry on a 3730 DNA Analyzer (Applied Biosystems Inc., Foster City, CA, USA) at the Cornell University Life Sciences Core Laboratories Center. 

### 2.5. Genotyping

Chromatograms were trimmed and read in SeqMan (DNASTAR Inc., Madison, WI, USA). Complete MLSSR types were assigned after determining the allele combinations on locus 1, locus 6, locus 8, locus 9, and locus 10. Locus 1 is a mononucleotide repeat, all others are trinucleotide repeats. All samples with greater than or equal to 11 G repeats at locus 1 were considered equivalent due to PCR amplicon slippage above 11 repeats. When there were multiple SSR types in a single sample, confirmed by homopolymer reads in both forward and reverse directions, both potential SSR types were assigned. When we were unable to distinguish the combinations of SSR types due to multiple loci with multiple detected alleles, the sample was assigned a ‘mixed’ designation. For these mixed samples, all possible permutations of two concurrent infections were considered as possible donations from the infectious dam. 

The subsets of fecal samples that were not collected during the scheduled semi-annual full-herd sampling were included in the evaluation of the number of SSR types per host and time to evolution events but were not used for calculation of herd MAP-infection incidence.

### 2.6. Data Deposition

Strain data, fecal shedding data and sampling information are stored in the Regional Dairy Quality Management Alliance (RDQMA) database, and samples from this study are preserved at University of Pennsylvania in the RDQMA BioBank. Contact the corresponding author for more information.

### 2.7. Data Analysis

SAS v 9.2 was used for all data management and statistical analysis (SAS Corporation, Cary, NC, USA). Figures were produced using GraphPad Prism (ver. 6, La Jolla, CA, USA). 

### 2.8. Vertical Transmission

The expected number of dam–daughter MAP-positive pairs was calculated by multiplying the proportion of positive dams, the proportion of positive daughters, and the total number of tested pairs. Dams with multiple daughters were included in each pair. The SSR types of all samples from each dam–daughter pair of MAP-positive animals were compared to determine whether the pair members shared the same SSRtype. An animal was considered to be MAP infected, or MAP positive, if any sample from the animal cultured positive for MAP, either during the live phase of the study or post-mortem. The distribution of within-host SSR types was calculated for all positive cows over the course of the study. The expected probability of vertical transmission of specific strains (independent of potential vertical transmission) was calculated assuming a Poisson distribution of positive counts in dams and daughters (Exact Poisson test). These results are referenced briefly in Mitchell et al. [20]. Vertical transmission was defined as an observed SSR type identity between dam and daughter and could be the result of either in-utero transmission, fecal-oral transmission in early life or infection from colostrum. When calculating the number of expected transmission events for individual SSR types, the expected number of MAP positive daughters was only calculated for the two dominant on-farm SSR types due to the low number of expected events for minor SSR types.

### 2.9. Evolution and Transmission Events

Following genotyping, a total repeat difference (RD) was assigned for each possible pair of SSR types within a sample or host. The RD was calculated as the sum of the absolute number of differences in repeats across all SSR loci. For cows with 3 SSR types over the course of the study, each pair of SSR types was treated separately. For example, a cow with three SSR types 7-5-6-5-5, 7-5-6-5-4 and 7-5-5-5-5, would be assigned 3 RD values of 1, 1, and 2. An expected repeat difference (ERD) was calculated by summing RD across all possible SSR type combinations weighted by observed SSR type frequency. The ERD reflects the probability of pairs with repeat difference (*n*) if selected at random from the population of all SSR types. ERD was also calculated at a cow level:
$$ERD\left(n\right)=\sum_{j=1}^{j=max}P\left(j\right)\sum_{\begin{array}{c} i:i\neq j\\ \delta \left(i,j\right)=n \end{array}}P\left(i\right)$$
where ERD(*n*) is the proportion of randomly selected pairs with an ERD of *n*, *j* is any strain, *P*(*j*) is the probability of any strain j being randomly selected from the population of all strains, *i* is any strain that is not *j* (with a repeat difference *δ* in the pair (*i*,*j*) being equal to *n*), and *P*(*i*) is the probability that strain *i* is selected randomly from the population of all strains in the population *i* ≠ *j*. 

Following calculation of sample-level and cow-level ERD, the observed distribution was compared to the expected via an exact binomial test with the ERD as the reference value. Provided that the observed RD within sample or host was significantly smaller than the ERD, we hypothesized that within-host evolution was likely.

### 2.10. Minimum Time-to-Mutation Estimate in Case of Strain Evolution

A Kaplan-Meier estimator was used to determine time to a single mutation at the SSR loci within a host. Animals with only one SSR type of MAP across all samples were assumed to represent baseline infection events with an ancestral SSR type. If animals had multiple SSR types with an RD of > 1, these SSR types were assumed to come from independent infection events. For animals with 3 SSR types, the SSR type that minimized individual repeat differences for a single pair was considered the potential ancestral strain. For animals infected concurrently with MAP SSR types that showed RD = 1, pairs were assumed to arise via within-host mutation. Multiple pairs within an animal were considered.

For the purpose of determining the maximum possible duration of infection prior to a mutation event, we assumed that all animals were infected at birth. Age at detection of a second closely-related SSR type (with one SSR type that was previously detected in the same host) allowed for the determination of maximum time to a mutation event. For animals with only one MAP SSR type and animals with multiple unrelated SSR types, maximum duration of infection was calculated from birth to the last sampling time point at which the MAP SSR type was detected. These animal-SSR type combinations were considered censored at the last available observation. 

The specific SSR locus at which a mutation occurred was not considered in the mutation-rate calculation. Because of the predominance of multiple infections with the two dominant on-farm SSR types, the possibility of multiple infection events rather than within-host mutations at this locus could not be ignored. To restrict the dataset to likely mutation events, animals with multiple infections of only the two dominant on-farm SSR types were initially excluded from the analysis. These samples were subsequently included to illustrate maximum possible mutation rate. These two Kaplan–Meier curves therefore represent the minimum and maximum rates of mutation if all infections occur at birth and all mutation events occur shortly before detection.

## 3. Results

The dairy herd in this study consisted of approximately 330 milking and dry cows and remained stable in terms of herd size during the years of the study.

### 3.1. Fecal Samples

In total, 4004 fecal samples were obtained from 1058 cows in this single dairy herd. The majority of these samples (3980) were collected on a semi-annual basis, while the remaining 24 were supplementary samples taken from known-positive or suspect animals. All 24 supplementary fecal samples were positive. From these 24 samples, SSR types were available from 22 samples. These 22 samples yielded the originally-recovered SSR type from the semi-annual sampling in addition to 4 samples with SSR types not previously detected. The semi-annual fecal samples were MAP positive in 1.8% of samples (77 out of 3980 samples), with an SSR type available for 69 out of these 77 MAP positive samples. In total, SSR types were available from 91 samples consisting of 69 from the semi-annual samples and 22 from the follow-up samples. The percent of cows with at least one positive result from a fecal sample was higher, 4.5%, 47 out of 1036 cows, since many cows contributed multiple samples in their lifetime. Among the positive cows, on average 2.5 samples per cow, including both fecal and tissue samples, were positive, ranging from 1 positive sample to 16 positive samples. Of the 47 positive cows, SSR types were available for 45. A single SSR type within animal was isolated from 34 animals (corresponding to 78 samples), while multiple SSR types within an animal were found in 11 animals (13 samples). See Table 1 for complete results.

### 3.2. Post-Mortem Samples

Over the course of the study, 775 post-mortem samples were collected from 155 cows. Of these cows, 67 cows (162 samples) were positive post-mortem in at least one sample type, corresponding to a percent positive of 43.2% at the cow level and 20.9% at the sample level. SSR types were available for 61 of these animals (with a total of 113 samples), with a single SSR type cultured from 100 samples of 48 animals and multiple SSR types cultured from 13 samples of 13 animals. These data are also summarized in Table 1. In Figure 1 the maximum colony forming units (CFU) in all fecal samples of a cow was plotted against maximum CFU of the post-mortem samples.

### 3.3. SSR Types

There were two dominant SSR types found on the study farm (MLSSR types 7-5-6-5-5 and 7-5-5-5-5) and at least 12 additional SSR types were identified (Figure 2). There were a total of 13 SSR types recovered from post-mortem samples and nine recovered from fecal samples. A percentage of 24% of all animals with a positive culture and SSR typing information available (23 out of 97) had more than one MAP SSR type recovered, with 10 animals presenting multiple SSR types in sequential samples and 19 animals presenting multiple SSR types detected concurrently. Here, sequential means the presence in multiple samples over time and concurrent indicates the presence of multiple SSR types at the same sampling time. Among the animals infected concurrently with multiple SSR types, about half showed multiple SSR types in the same sample (12), while the other half showed multiple SSR types in multiple samples collected at the same point in time (12) (see Table 1).

### 3.4. Dam-Daughter Pai

We identified 736 dam-daughter pairs with at least one pair of fecal or post-mortem culture results available in the herd. Of 50 daughters that were MAP infected in these data, 13 were from a known MAP infected dam (26%). Taking the available MAP SSR type information into account, 9 out of 50 MAP infected daughters had a dam infected with the same MAP SSR type (18%). The results of an exact Poisson test revealed a higher number of observed dam-daughter positive pairs than expected under assumption of random MAP infections (13 vs. 8.5 pairs, respectively, *p*-value = 0.05). When fecal samples alone were considered, there were also significantly more positive dam-daughter pairs than expected (5 vs. 1.3, *p* < 0.01). 

Of the 12 MAP-shedding dam-daughter pairs with MAP SSR type information available (one pair had incomplete SSR typing data), nine shared the same MAP SSR type, and two of these nine daughters also carried an additional MAP SSR type with a repeat difference (RD) of 1 relative to the shared SSR type. In three pairs, daughter SSR types did not match any of the SSR types identified in the dam, in one of these three, an SSR type with an RD of 2 was present in the infected daughter relative to an SSR type in her dam. All dam–daughter pairs with a common MAP SSR type shared one of the two dominant SSR types (see Table 2). The number of dam–daughter pairs infected with 7-5-6-5-5 was significantly higher than expected (8 vs. 3.4, *p* = 0.02). On the other hand, the number of pairs infected with 7-5-5-5-5 was not significantly different than expected (3 observed vs. 1.2 expected, *p* = 0.12). 

### 3.5. Co-Infections

Of the 26 samples with multiple SSR types, 22 comprised pairs of the two dominant on-farm SSR types (7-5-6-5-5 and 7-5-5-5-5) with a repeat difference (RD) of 1. Of the remaining four pairs with non-dominant strains, three also had an RD = 1, and one pair had an RD = 4. The expected repeat difference (ERD) between different single SSR type samples was significantly higher than the observed RD within multiple-SSR type samples (Exact test for binomial proportion: RD = 1 vs. RD > 1, *p* < 0.001) (see Figure 3A). 

At the cow level, there were 74 animals with a single MAP SSR type recovered and 23 cows (24%) with multiple SSR types recovered. The distribution of all pairs of SSR types in these cows relative to the ERD showed a significantly higher observed frequency of a single repeat difference relative to the expected number of single repeat differences. (Exact test for binomial proportion: RD = 1 vs. RD > 1, *p* = 0.005). (Figure 3A).

In seven animals there were multiple SSR types recovered that had an RD of ≥ 2. There were six animals with concurrent SSR types (recovered from the same sample or from different concurrently-collected post-mortem samples) that were not pairs of the dominant on-farm SSR types and were within one RD of each other (Table 3). In these six animals, one had three closely related SSR types recovered at post-mortem (see Table 3). Because SSR type 7-4-6-3-5 was found in just one animal in the herd, it was assumed that the two-mutation change was a result of two sequential evolution events rather than a separate infection event.

### 3.6. MAP Evolution Rate Estimation

We estimated that within-host evolution rates of SSR regions in MAP SSR types averaged eight years (95% CI: 7.1–9.0 years) if all co-infections with the two dominant MAP SSR types are considered to be single transmission events at birth. For co-infections with non-dominant SSR types, the mean estimate of within-host evolution rate increased to 9.9 years (95% CI: 8.7–11 years). 

## 4. Discussion

### 4.1. The Majority of True MAP Infections Are Likely Not Detected by Fecal Culture 

In our study, MAP prevalence was demonstrably higher in the post-mortem samples compared to the on-farm fecal samples, with 20.5% prevalence in post-mortem culture and 1.8% prevalence in fecal culture. Culling decisions were predominantly based on fertility, low production, lameness, or mastitis. No additional culling was done based solely on MAP results. It can be thus inferred that a large proportion of true MAP infections are not detected by fecal culture when evaluated semi-annually. Indeed, of the 148 animals with post-mortem results available, only one cow with less than 150 CFU/tube at post-mortem shed MAP in a previous fecal sample (see Figure 1). These data suggest a potential infection threshold that must be reached in the tissue before a fecal sample will become culture positive. The longitudinal, repeated-measures data presented in the current study indicate that even semi-annual fecal culture schemes may be ineffective at successfully identifying the majority of MAP infected animals in a herd. These results imply that the sensitivity of fecal culture is likely lower than traditionally reported [16]. This low observed sensitivity of fecal culture will have a major impact on our understanding of the efficacy of control programs relying exclusively on fecal culture.

### 4.2. Multiple Strain Types (Co-Infections) within a Single Host Are Common

The presence of multiple SSR types within a single host is not infrequent: approximately 24% of all animals (23 out of 97) with an SSR positive culture, where SSR typing information was available, had more than one MAP SSR type recovered during the study. Multiple SSR types of MAP were observed in different samples from a given animal, but also within a single sample. In future work, it would be of particular interest to determine if co-infection with more than one MAP SSR type predisposes animals to shedding or an accelerated clinical disease progression. Several studies on other mycobacterial species support the plausibility of this hypothesis. For instance, superinfecting *M. marinum* were shown to travel into previously-established granulomas and rapidly adapt to long-term survival; thus, pre-existing granulomas are evidently unsuccessful at eliminating new mycobacteria previously unexposed to host immune response [21]. In *M. tuberculosis*, multi-strain co-infection during the infectious stage were reported to impact and advance the evolutionary strategies of the bacteria as evaluated in mathematical models [22]. 

Co-infections with two MAP strains have been identified in previous work on MAP-infected cattle [12,13], but there have been relatively few studies evaluating this phenomenon. In contrast, there is more information regarding co-infection with two *M. tuberculosis* strains in human hosts [23,24]. The prevalence of *M. tuberculosis* co-infections (10–20%) [22], is comparable to the prevalence of MAP co-infections observed in the present study (24%). With respect to *M. tuberculosis*, co-infections are increasingly observed in high tuberculosis-incidence communities and high-population-density settings [23,24], suggesting a connection between density of contacts and the development of co-infections. In cattle, co-infections with certain pathogens have been shown to depress immune function and enhance shedding of other pathogens [25]; plausibly, such co-infections could influence susceptibility to multi-strain MAP infections and impact shedding patterns.

### 4.3. Vertical Transmission Appears Present in Positive Dam–Daughter Pairs

Vertical transmission appears to be a possible pathway in 10 out of 50 infected daughters (Table 4 and Table 5). If only fecal samples were considered, as consistent with previous work [17], there were significantly more MAP-positive dam–daughter pairs in our study than expected through random horizontal transmission alone (Table 4). When assuming that within-host mutations could account for single repeat differences (and one two-repeat difference), vertical transmission was likely in 10 of 12 infected dam–daughter pairs with MAP SSR data. In one of these 10 pairs (pair 2 in Table 5), the dam and daughter did not share a common SSR types but only SSR types with an RD of 2. This raises the possibility of strain replacement after the mutation event. Replacement has been hypothesized in other mycobacterial species, such as the opportunistic *M. xenopi* [26], but is more frequently documented in pathogens such as *M. tuberculosis*, when drug resistance influences fitness and within-host strain diversity [22]. In reference to MAP, Dennis et al. [27] investigated clinic-pathological features of Johne’s disease in infected sheep and observed, based upon extensive post-mortem culture, that a portion of the animals (8.7%) appeared to clear the infection entirely. Thus, the potential for an animal to recover from (or fail to sustain) an infection with one MAP strain while maintaining a mutated strain is reasonable, though such an occurrence is likely rare.

Interestingly, all dam–daughter pairs with a common MAP SSR type shed at least one of the two dominant SSR types on the farm. These two SSR types comprised more than half of the shedding on the farm, as well as more than half of all infected animals as assessed by post-mortem sampling (see also Table 2). Despite the additional risk of contracting these dominant SSR types through horizontal transmission events, the number of dam–daughter pairs with the same SSR type was significantly higher than expected for SSR type 7-5-6-5-5 but not for 7-5-5-5-5. 

### 4.4. Vertical Transmission Is Not the Dominant Transmission Pathway Overall

The observed number of MAP-positive dam-daughter pairs in this cohort is larger than what would be expected by horizontal transmission alone. Still, this route of transmission has a limited contribution to overall MAP infection dynamics. In only 10 of the 125 dam–daughter pairs with at least the dam MAP infected and with SSR data available, an apparent MAP transmission event from dam to daughter took place. Additionally, while vertical transmission is likely responsible for many of the MAP-positive pairs observed, daughters also have SSR types that are commonly found in the farm environment and are not necessarily transmitted from the dam. In closed herds, the unique daughter strains may represent new introductions of MAP from external sources such as wildlife, people or transportation vehicles. Alternatively, the appearance of apparently new strains may be attributed to within-host mutation or to infections that are not detected by fecal culture and only detectable post mortem.

It should be noted that by the very nature of the dam-daughter relationship, the dam has a longer time on the farm and in the study compared to the daughter. This means that the detection opportunity for dams is higher compared to daughters. It also means that in calendar time, the daughters were born later compared to the dams and the awareness and subsequent preventative management of the farm staff with regard to MAP may have increased over the course of the study. Both reasons would result in a lower MAP infection risk in daughters compared to dams, this lower risk is clearly visible in the results presented in Table 4.

Our results are consistent with estimates of vertical transmission in current mathematical models; Mitchell et al. [28] concluded that the contribution of vertical transmission did not explain MAP persistence following implementation of control strategies. Thus, as concluded by Mitchell et al. [28] vertical transmission is not a dominant transmission pathway and would be insufficient by itself to maintain infection. Yet, vertical transmission was shown to impact the spread of MAP when combined with transient shedding by calves and heifers, suggesting that this transmission route may increase incidence of new MAP infections and impact the spread of MAP under certain conditions.

### 4.5. Within-Host Evolution

Distinguishing within-host evolution from the introduction of new strains becomes critical when evaluating strain dynamics within a given herd. In the multiple strain samples in our study, the expected number of pairs with a repeat difference of more than 3 is larger than the observed number of pairs with a repeat difference of more than 3 (Figure 3), suggesting that within-host evolution likely explains a portion of the observed MAP co-infections. We calculated an average within-host mutation rate of the SSR regions within the MAP genome of approximately eight years, assuming all co-infections with the two dominant MAP SSR types are considered single transmission events at birth. The evolutionary rates for SSR loci in MAP have not previously been estimated; however corresponding rates are also low for other mycobacterial species. For example, researchers have estimated a molecular clock of 0.3–0.5 mutations/genome/year in *M. tuberculosis* [29]. In *M. bovis*, a rate of 0.147 substitutions per genome per year was presented [30]. It is certainly conceivable that the mutation rate for MAP may be lower, in part owing to a slower generation time for MAP compared to other mycobacterial species [31]. In any case, the current observed mutation rate of eight years (0.125 substitutions per genome per year) appears to be in reasonable agreement with the approximate 7 year rate observed in *M. bovis* [30]. It should be noted however that the expansion and contraction of SSR regions in the genome are largely attributed to strand slippage during DNA synthesis [32]. The propensity of SSRs to mutate increases with their repeat number, likely reflecting the increased probability of strand slippage with length [32]. Additionally, SSR loci may be exposed to higher selection pressures as these loci are often located in regulatory regions. Studies have shown that starting at a certain repeat number, SSRs can acquire mutation rates greater than those of non-repetitive loci and/or loci with just two repeats [33]. Therefore the observed mutation rate in SSR regions may overestimate the overall mutation rate in the bacterial genome.

The productive lifespan of a U.S. dairy cow is decreasing and was recently estimated at 2.63 years, corresponding to a total lifespan of 4.8 years, on average. Therefore, within-host SSR mutations for MAP are likely not undergoing selection at a rate faster than the lifespan of a dairy cow. Additionally, barring specific selective pressure on a single SSR locus, it is unrealistic that the vast majority of mutation events would be repeated mutations from one dominant type to the other; therefore, the number of mutation events, if anything, is overestimated. 

We performed our study in a single dairy herd. This dairy herd has been followed over approximately 10 years with more or less continuous bi-annual fecal sampling. The herd staff were also very meticulous in recording cow data and were very willing to perform the additional work necessary for a study of this type. The farm was essentially a closed operation with no animals being purchased from the outside. The advantage of this farm is the precise and longitudinal follow up of MAP infections in this population. The limitation of using data from this one herd is the limited ability to easily expand the results of this study to all dairy herds. Hence, the results of this study should be interpreted with care. On the other hand, there are very few, if any, data sets available with the precision and completeness as the data collected in this longitudinal study. The biological phenomena that we are observing in this population may therefore still have external validity and help our further understanding of MAP epidemiology and patho-biology.

Our analyses were based on the assumption that when a smaller within-host RD was observed compared to the expected RD across all hosts, this small within-host RD was the result of a mutation in the DNA of the strain and not the result of a second infection event. It is clear that we do not have full certainty that these assumptions will be correct in all events. However, the use of similar sequence data, including SSR results allowed Biek et al. [30] to develop hypotheses on transmission of *M. bovis* within and across herds and transmission between cattle and badger populations. We would argue that the results of this study certainly give rise to the hypotheses that in case of multiple SSR strains recovered from the same host, within-host mutations are to be more likely compared to a second new infection event. Further analyses of these data, may allow further detailed understanding of MAP transmission patterns within a closed population. Our data were based on detailed sequence analyses of five short sequence repeat regions in the genome. The use of a limited number of short sequence repeat regions for strain typing and subsequent mutation analyses has its limitations compared to whole genome sequencing. On the other hand, these SSR based methods are readily available for research purposes and the results of this study will therefore have a large external use value compared to the much more complex and expensive whole genome sequencing. It is expected that future studies on this elaborate dataset will incorporate the use of whole genome sequencing. Such results will then be reported and compared to the results reported in this publication. 

## 5. Conclusions

We set out to make a plausible distinction between horizontal and vertical transmission events and to discern multiple infection events from within-host evolution in a longitudinal MAP cohort. MLSSR-based typing provides an invaluable method for differentiating multiple infections that is not possible in band-based typing systems. The longitudinal sampling scheme for the current study provides key advantages over cross-sectional data using the same MLSSR typing scheme, as more MAP strains were identified in this study than in several cross-sectional studies with similar methodology [12,13]. 

A high proportion of MAP infected cows with multi-strain MAP infections were identified, most of which appeared to result from multiple infection events; however, within-host mutation is the likely explanation for at least a portion of these observed co-infections. Our research shows a strong association between within-host multiple-SSR type recovery and small repeat-difference values of the recovered strain pairs (Figure 3A). We have provided an estimate of approximately 8 year for the time-to-mutation for the SSR region of MAP strains in the study. 

Regarding the importance of vertical transmission, approximately twenty percent of fecal and post-mortem culture-positive daughters likely resulted from their dams being infected with the same MAP strain. However, only ten percent of infected dams transmitted MAP to their daughters. Thus, vertical transmission may play a role in approximately twenty percent of infected daughters from infected dams. Finally, we have concluded that even a semi-annual fecal sampling scheme is vastly insufficient to classify the majority of infected animals on a farm. Post-mortem sampling is an important tool for accurately determining the infection status of the animal due to a potential burden that must be reached in the tissue before a fecal sample will become culture positive. Our precisely-documented, longitudinal dataset has provided improved insight into MAP transmission events, infection pathways, and within-host mutation. This insight may ultimately contribute to the control of mycobacterial co-infections.

## Figures and Tables

**Figure 1 vetsci-06-00032-f001:**
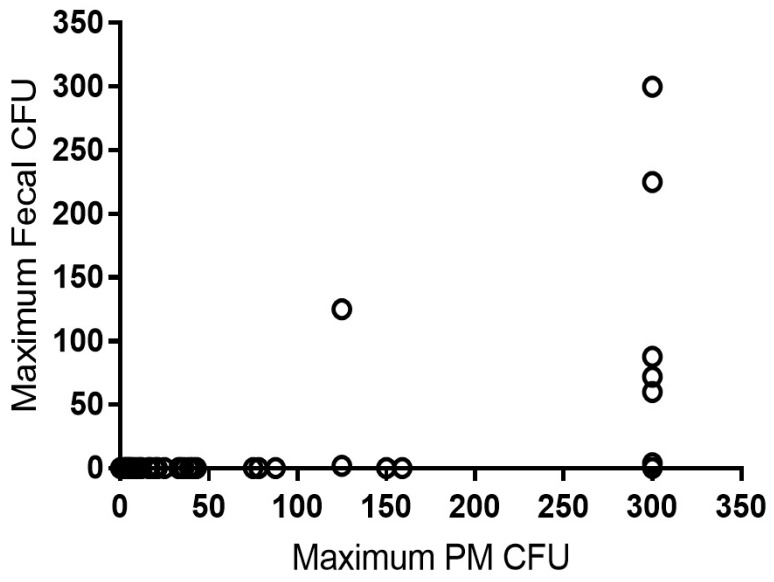
Distribution of colony forming units among samples collected from cows on a single dairy farm. The maximum post-mortem colony forming units (PM CFU) per tube is shown on the *X* axis and the corresponding maximum lifetime fecal CFU per tube in the same cow is shown on the *Y* axis.

**Figure 2 vetsci-06-00032-f002:**
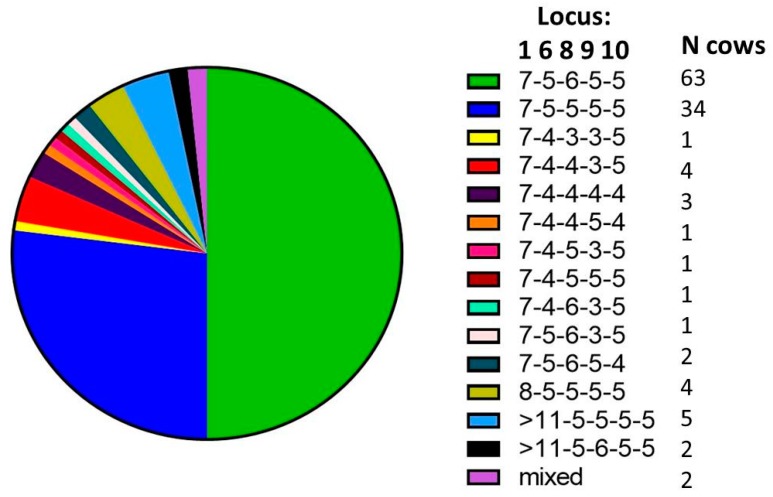
Distribution of Short Sequence Repeat Typing (SSR) types identified over the course of seven years on the study farm. Data at cow level, 124 SSR types in a total of 97 samples.

**Figure 3 vetsci-06-00032-f003:**
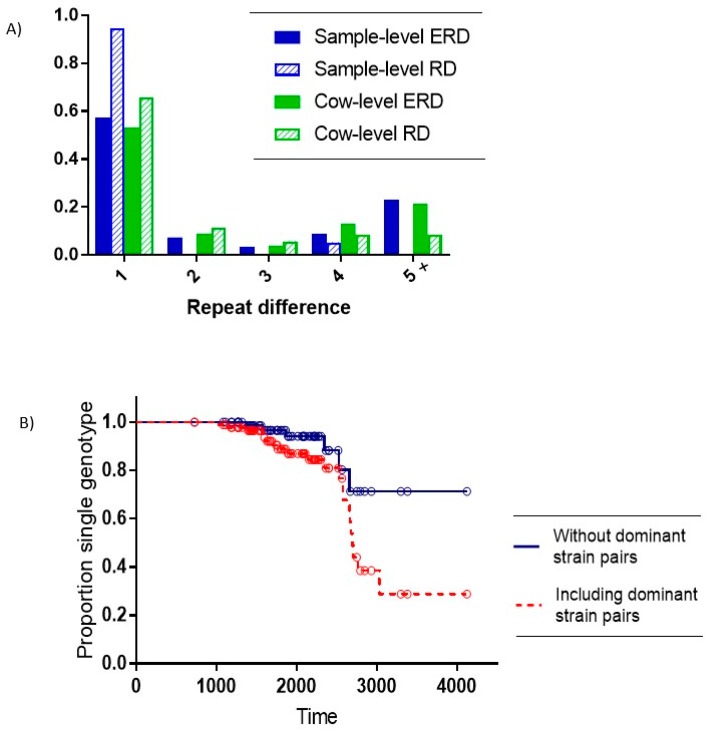
Distinguishing multiple infections from within-host evolution. (**A**) Juxtaposes the Expected Repeat Difference (ERD) based upon random pairs of single-SSR type samples with the observed Repeat Difference (RD) for within-host pairs with multiple SSR types. Solid bars represent ERD and striped bars represent RD. Sample-level results are presented in blue and cow level results in green. The *X* axis shows repeat differences of 1, 2, 3, 4, and 5 or more. The *Y*-axis indicates the proportion of samples. The four samples with ‘mixed’ designation are not included in this figure. (**B**) Kaplan–Meier plot of time in days to mutation event in animals with at least one SSR type initially detected. The red dashed line includes dominant on-farm SSR pairs, while the solid blue line does not.

**Table 1 vetsci-06-00032-t001:** Number of fecal and post-mortem cultures processed, summarized at both sample and cow level. Fecal samples from semi-annual (SA) sampling and supplementary samples (SS) from known shedders are shown. Post mortem (PM) samples were collected after slaughter in the slaughter house. Multiple SSR types are divided into sequential time-points or concurrent samples (further categorized by their presence in the same or separate samples).

	Sample Level			Cow Level		
	Fecal		PM	Fecal	PM	Total *
	SA	SS				
Total	3980	24	775	1036	155	1058
Positive	77	24	162	47	67	103
SSR type(s) available	69	22	113	45	61	97
Single SSR type	60	18	100	34	48	74
Multiple SSR types	9	4	13	11	13	23
Sequentially	-	-	-	8	-	10
Concurrently	9	4	13	8	13	19
In same sample	9	4	13	8	9	12
In different samples	-	-	-	-	12	12

* This column does not add up to the sum of the two adjacent columns because of some overlap at the cow-level between the two sample types.

**Table 2 vetsci-06-00032-t002:** SSR types of *Mycobacterium avium* subsp. *paratuberculosis* (MAP) summarized at sample level and at cow level. Samples are presented from semi-annual (SA) and supplementary sampling (SS) as well as post-mortem (PM). The number of samples with multiple SSR types is also provided. At cow level, some samples have more than two SSR types represented. For a complete breakdown of SSR types present on the farm see Figure 2.

SSR Types	Samples ^1^				Cows		
	Fecal	SA	SS	PM	Fecal	PM	Total Positive
All SRR types available	91	69	22	113	45	61	97
7-5-6-5-5	68	50	18	75	32	40	64
7-5-5-5-5	26	20	6	22	18	19	35
Other	11	9	2	23	7	15	22
Multiple SRR types	13	9	4	13	11	13	23

**^1^** Total individuals positive is broken down into the two dominant MAP SSR types and ‘Other’ low frequency SSR types present on the farm.

**Table 3 vetsci-06-00032-t003:** Animals with concurrent SSR types recovered where RD = 1. N is number of likely evolution events. Days is the age of the cow in days at sampling where the isolates were obtained. Type is the sample from which the SSR types were recovered, fecal (F) or post-mortem (PM).

*n*	Cow	Birthdate	Sampling	Days	Type	SSR Type 1	SSR Type 2	SSR Type 3
1 **	1178	7/Oct/1999	11/Nov/2004	1862	PM	7-5-6-5-5	7-5-6-5-4	
1	1231	16/Nov/2000	15/Oct/2007	2524	F	7-5-5-5-5	8-5-5-5-5	
1 **	1300	18/Jul/2001	28/Oct/2008	2659	M	7-5-6-5-4	7-5-6-5-5	
1	1416	4/Dec/1999	01/May/2006	2343	F	7-4-4-4-4	7-4-4-5-4	
3 **	1751	4/Aug/2004	03/May/2008	1368	PM	7-4-3-3-5	7-4-4-3-5	7-4-6-3-5
1	1683	11/Jan/2004	03/May/2008	1574	PM	7-4-4-3-5	7-4-5-3-5	

** Represent concurrent samples from different sites at post-mortem.

**Table 4 vetsci-06-00032-t004:** Dam–daughter pairs with MAP culture results in both dam and daughter. Dams may have multiple daughters and are in such cases represented in multiple pairs. Overall MAP status based on all SSR types and results for individual SSR types 7-5-6-5-5 and 7-5-5-5-5 include both fecal and post-mortem results. MAP infection status based solely on fecal culture and MAP infection status solely based on post-mortem culture are for all strains. Results shown are observed and (expected) cases. A significant difference between observed and expected is indicated by an asterisk *.

Map Source	Dam:	Infected	Infected	Not Infected	Not Infected	Total
	Daughter:	Infected	Not Infected	Infected	Not Infected	
All cultures, all SSR types		13 (8.5) *	112	37	574	736
All cultures, 75,655		8 (3.4) *	70	24	634	736
All cultures, 75,555		3 (1.2) *	51	14	668	736
Fecal, all SSR types		5 (1.3) *	56	11	659	731
Post-mortem, all SSR types		5 (5.2) *	9	1	4	19

**Table 5 vetsci-06-00032-t005:** MAP SSR types from bacterial isolates identified from samples taken from animals in 12 daughter and dam pairs where both daughter and dam were MAP infected. Sample type, post-mortem (PM) or fecal, is indicated for samples from daughters and dams. Any SSR type that is common to dam and daughter is denoted in the same horizontal line.

Pair	Dam SSR Type	Daughter SSR Type	Possible Direct Dam-Daughter Transmission
1	7-5-5-5-5 (Fecal)	- *	No
	-	7-4-4-3-5 (PM)	No
	-	7-4-6-3-5 (PM)	No
	-	7-4-3-3-5 (PM)	No
2	8-5-5-5-5 (PM)	-	No
	7-4-4-4-4 (PM)	-	No
	7-4-5-5-5 (PM)	-	No
	-	7-4-4-3-5 (Fecal, PM)	No
	-	7-4-5-3-5 (PM)	No
	-	7-5-6-5-5 (PM)	No
3	7-5-6-5-5 (PM)	7-5-6-5-5 (Fecal)	Yes
4	7-5-6-5-5 (Fecal, PM)	7-5-6-5-5 (Fecal)	Yes
	-	7-5-5-5-5 (PM)	No
5	7-5-6-5-5 (Fecal)	7-5-6-5-5 (Fecal, PM)	Yes
	7-5-5-5-5 (Fecal)		No
6	** 7-(4/5)-(5/6)-5-5	7-5-5-5-5 (PM)	Yes
	7-(4/5)-(5/6)-5-(4/5)		No
	7-(4/5)-(4/5)-(5/4)-(4/5)		No
7	7-5-6-5-5 (PM)	7-5-6-5-5 (PM)	Yes
8	7-5-6-5-5 (Fecal)	7-5-6-5-5 (Fecal)	Yes
	7-5-5-5-5 (Fecal)		No
9	7-5-5-5-5 (PM)	7-5-5-5-5 (Fecal, PM)	Yes
	7-5-6-5-5 (Fecal, PM)	7-5-6-5-5 (PM)	Yes
	-	7-5-6-5-4 (PM)	No
10	-	7-5-6-5-5 (PM)	No
	7-4-4-4-4 (PM)	-	No
11	7-5-6-5-5 (PM)	7-5-6-5-5 (PM)	Yes
	7-5-5-5-5 (PM)	7-5-5-5-5 (PM)	Yes
12	7-5-6-5-5 (Fecal)	7-5-6-5-5 (Fecal)	Yes
	7-5-5-5-5 (Fecal)		No

* Means same SSR type not found in either dam or daughter; ** Because this SSR type could be isolated from the dam’s multiple ‘mixed’ types, we interpret it as a possible shared SSR type.

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
