# Peer review of "Elucidating Transmission Patterns of Endemic Mycobacterium avium subsp. paratuberculosis Using Molecular Epidemiology"

_vetsci, 2019, doi:10.3390/vetsci6010032_

Round 1
Reviewer 1 Report
The group of scientists conducted analysis of molecular epidemiology of MAP using data from a very well designed longitudinal study on a dairy farm. The study provides insights into impact of vertical transmission on disease transmission, actual fecal testing sensitivity, and within host evolution of MAP. All of these would be of great interest of scientists in the field and would help advancement of research in the field. The manuscript is well organized and presented their finding very well. I have only minor comments as follows.
Overall: I may have missed but vertical transmission in this manuscript includes all of in utero, fecal-oral in maternity barn and infection from colostrums?
Line 125: Please elaborate on “CFU ranges based on colony density rather than individual count”. To me it was not clear.
Line 129: It is stated that colonies were preserved on beads – please provide a little more details of this procedure. I could not find it in references 17-19.
Paragraph 201-224: Some spacing missed.
Table 1 legend: Explain what the following mean - Sequentially, Concurrently, In same sample, and In different samples.
Table 2: Contents are not aligned.
Line 415: Where the “10 out 50” came from?
Table 4: Unnecessary line before the number “56”
Table 5: “Evidence” for direct dam-daughter transmission may be overstatement.
Author Response
The group of scientists conducted analysis of molecular epidemiology of MAP using data from a very well designed longitudinal study on a dairy farm. The study provides insights into impact of vertical transmission on disease transmission, actual fecal testing sensitivity, and within host evolution of MAP. All of these would be of great interest of scientists in the field and would help advancement of research in the field. The manuscript is well organized and presented their finding very well. I have only minor comments as follows.
Overall: I may have missed but vertical transmission in this manuscript includes all of in utero, fecal-oral in maternity barn and infection from colostrums?
AU: Correct, this was clarified now in the manuscript [line 213]
Line 125: Please elaborate on “CFU ranges based on colony density rather than individual count”. To me it was not clear.
AU: The description is indeed not very clear. Changed to: samples with high counts (>75) were assigned CFU ranges based on colony count estimates rather than precise individual counts.
Line 129: It is stated that colonies were preserved on beads – please provide a little more details of this procedure. I could not find it in references 17-19.
AU: clarified in the text as: colonies from that tube were preserved on porous beads, which are intended as carriers to support the viability of microorganisms during storage in the freezer. A single frozen bead was then used to re-inoculate a HEYM tube, representing a third generation of culture.
Paragraph 201-224: Some spacing missed.
AU: changed.
Table 1 legend: Explain what the following mean - Sequentially, Concurrently, In same sample, and In different samples.
AU: this was now further explained in the text: Here, sequential means the presence in multiple samples over time and concurrent indicates the presence of multiple SSR types at the same sampling time. Among the animals infected concurrently with multiple SSR types, about half showed multiple SSR types in the same sample (12), while the other half showed multiple SSR types in multiple samples collected at the same point in time (12) (see table 1).
Table 2: Contents are not aligned.
AU: changed.
Line 415: Where the “10 out 50” came from?
AU: 10 are likely vertical transmission events [table 5], 50 are the infected daughters with a dam that has a known MAP infection status [table 4]. This has been clarified in the text.
Table 4: Unnecessary line before the number “56”
AU: removed
Table 5: “Evidence” for direct dam-daughter transmission may be overstatement.
AU: changed to ‘Possible Dam Daughter transmission’
Reviewer 2 Report
This manuscript describes a very strong dataset of longitudinally sampled cows (via fecal samples and post-mortem tissue samples), including dams and daughters. The authors used short sequence repeat typing to differentiate strains and tried to decipher within host mutation of SSR loci versus secondary infection with a different strain. The authors also attempted to explain the role of vertical transmission from dam to daughter versus horizontal transmission. The manuscript is well written and most of my comments are minor. My main concern is that I was not able to understand the co-infection results (ERD and RD) and thus the conclusions derived from them.
Line 96: During which years was sample collection performed?
Line 106: Were producers made aware of which cows had MAP-positive fecal samples?
Lines 187–188: Suggest adding the type of repeat for each of the 5 loci used in this study (e.g. monnucleotide or trinucleotide repeats).
Line 284: Were there any instances of MAP-positive fecal shedding with a negative post-mortem culture?
Line 298: It would be interesting to see SSR types over time. Were all SSR types found throughout the 7 year period, or did some appear later in the study?
Line 300: Figure 2 caption – Would be nice to indicate which loci (i.e. 1-6-8-9-10) are presented. It would also be useful to show the number of isolates found with each SSR type (either at cow or sample level or both) next to the legend.
Lines 341–354 (and discussion of these results elsewhere in the manuscript): This section is unclear.
· The authors mention “mixed-SSR type samples” – does this mean mixed as defined at Line 192 or mixed as in multiple SSR types found in a single sample/cow. The legend in Figure 3 indicates that “mixed” designation were not included, so I assume it’s the latter, but consider rephrasing so this is clear.
· The Y-axis is not labeled in Figure 3A, though the caption defines it as what appears to be the X-axis label. It’s unclear what the Y-axis represents.
· The ERD appears to be smaller than the observed RD in Figure 3 with repeats 1–3, though in the text it is stated the ERD is larger than the observed.
· When the authors state “the expected repeat difference was significantly higher”, do the authors mean the probability of pairs with a repeat difference (n) is significantly higher than expected?
Line 346: Include unit for time (days?).
Line 379: Why were cows culled? Is the prevalence estimate based on post-mortem culture biased because MAP-positive animals were more likely to be culled?
Line 383: Figure 1 doesn’t show a fecal sample positive with post-mortem CFU <100. Do the authors mean <150 CFU?
Line 406: I was not able to find the prevalence of M. tuberculosis coinfections in the included citation (#23). Prevalence of coinfections will depend on the strain typing method used to decipher different strains, so I wanted to see if this was an appropriate “apples to apples” comparison. Please clarify.
Lines 455–459: Again, it would be interesting to see the SSR types over time to see if introduction during the course of the study is a plausible explanation.
Lines 483–488: This comparison is not appropriate – SSRs are specifically used to type monomorphic organisms because they are hypermutable. The 0.125 substitutions per genome per year is misleading, as in the current study substitution is a repeat unit (i.e. a 3-nucleotide unit for several of the loci). Comparing mutation rates of 5 short (potentially hypermutable) loci to mutation rates of SNPs from whole genomes is not a fair comparison, even with the explanation provided in Lines 488–494. I encourage the authors to remove these lines or rephrase to compare current results to SSR or VNTR loci in other organisms.
Line 498: Consider discussing the possibility that there are selective pressures at SSR loci, as this has been suggested extensively in the literature and SSRs are often located in regulatory regions.
Lines 500–502: This seems vague. Can the authors further explain how a strain-specific mathematical model or control program would be modified given the long latency?
Lines 528–530: Yes please! This would be a very strong dataset to explore the evolution and epidemiology of MAP over time using whole genome sequencing.
Author Response
This manuscript describes a very strong dataset of longitudinally sampled cows (via fecal samples and post-mortem tissue samples), including dams and daughters. The authors used short sequence repeat typing to differentiate strains and tried to decipher within host mutation of SSR loci versus secondary infection with a different strain. The authors also attempted to explain the role of vertical transmission from dam to daughter versus horizontal transmission. The manuscript is well written and most of my comments are minor. My main concern is that I was not able to understand the co-infection results (ERD and RD) and thus the conclusions derived from them.
Line 96: During which years was sample collection performed?
AU: Data were collected from 2004 to 2011. Added to text.
Line 106: Were producers made aware of which cows had MAP-positive fecal samples?
AU: Yes, this has been added to the text.
Lines 187–188: Suggest adding the type of repeat for each of the 5 loci used in this study (e.g. mononucleotide or trinucleotide repeats).
AU: Locus 1 is a mononucleotide repeat, all others are trinucleotide repeats. This was added to the text.
Line 284: Were there any instances of MAP-positive fecal shedding with a negative post-mortem culture?
AU: As can be seen from Figure 1, all cows with a positive fecal culture were also positive in the post-mortem samples.
Line 298: It would be interesting to see SSR types over time. Were all SSR types found throughout the 7 year period, or did some appear later in the study?
AU: Some of the SSR types were only identified later in the study. We decided not to include this as the study, although relatively long compared to other field studies, only reflects a few years in the total lifespan of a herd. So conclusion on new introductions would not be very strong.
Line 300: Figure 2 caption – Would be nice to indicate which loci (i.e. 1-6-8-9-10) are presented. It would also be useful to show the number of isolates found with each SSR type (either at cow or sample level or both) next to the legend.
AU: figure has been changed as suggested.
Lines 341–354 (and discussion of these results elsewhere in the manuscript): This section is unclear.
AU: Text has been edited to improve clarity.
The authors mention “mixed-SSR type samples” – does this mean mixed as defined at Line 192 or mixed as in multiple SSR types found in a single sample/cow. The legend in Figure 3 indicates that “mixed” designation were not included, so I assume it’s the latter, but consider rephrasing so this is clear.
AU: What is meant with ‘mixed’ is the definition as used in line 192. In the manuscript this was indeed not consistently used, and is now changed throughout.
The Y-axis is not labeled in Figure 3A, though the caption defines it as what appears to be the X-axis label. It’s unclear what the Y-axis represents.
AU: Correct, this was wrongly labelled. Now changed.
The ERD appears to be smaller than the observed RD in Figure 3 with repeats 1–3, though in the text it is stated the ERD is larger than the observed.
AU: This is now stated more clearly: The distribution of all pairs of SSR types in these cows relative to the ERD showed a significantly higher observed frequency of a single repeat difference relative to the expected number of single repeat differences. (Exact test for binomial proportion: RD=1 vs RD>1, P=0.005). (Figure 3, panel A).
When the authors state “the expected repeat difference was significantly higher”, do the authors mean the probability of pairs with a repeat difference (n) is significantly higher than expected?
AU: What is meant here is that the expected number of pairs with a repeat difference of more than 3 is larger than the observed number of pairs with a repeat difference of more than 3. This has been changed in the text.
Line 346: Include unit for time (days?).
AU: correct, it is days. Added to the figure.
Line 379: Why were cows culled? Is the prevalence estimate based on post-mortem culture biased because MAP-positive animals were more likely to be culled?
AU: Culling was done based on fertility, low production, lameness or mastitis. No additional culling was done based solely on MAP result. This has been added to the text.
Line 383: Figure 1 doesn’t show a fecal sample positive with post-mortem CFU <100. Do the authors mean <150 CFU?
AU: Yes, correct. Changed in the text.
Line 406: I was not able to find the prevalence of M. tuberculosis coinfections in the included citation (#23). Prevalence of coinfections will depend on the strain typing method used to decipher different strains, so I wanted to see if this was an appropriate “apples to apples” comparison. Please clarify.
AU: Correct, the citated reference uses mathematical modelling to come to their conclusion on evolutionary strategies. We clarified this in the paper. A reference on multiple infections in M. tuberculosis would be:
Cohen T, van Helden PD, Wilson D, Colijn C, McLaughlin MM, Abubakar I, Warren RM. Mixed-strain mycobacterium tuberculosis infections and the implications for tuberculosis treatment and control. Clin Microbiol Rev. 2012 Oct;25(4):708-19. doi: 10.1128/CMR.00021-12. Review
Lines 455–459: Again, it would be interesting to see the SSR types over time to see if introduction during the course of the study is a plausible explanation.
AU: We decided not to include this as the study, although relatively long compared to other field studies, only reflects a few years in the total lifespan of a herd. So conclusion on new introductions would not be very strong.
Lines 483–488: This comparison is not appropriate – SSRs are specifically used to type monomorphic organisms because they are hypermutable. The 0.125 substitutions per genome per year is misleading, as in the current study substitution is a repeat unit (i.e. a 3-nucleotide unit for several of the loci). Comparing mutation rates of 5 short (potentially hypermutable) loci to mutation rates of SNPs from whole genomes is not a fair comparison, even with the explanation provided in Lines 488–494. I encourage the authors to remove these lines or rephrase to compare current results to SSR or VNTR loci in other organisms.
AU: We agree with this point. Text was added to reflect this: It should be noted however that the expansion and contraction of SSR regions in the genome are largely attributed to strand slippage during DNA synthesis [33]. The propensity of SSRs to mutate increases with their repeat number, likely reflecting the increased probability of strand slippage with length [33]. Additionally, SSR loci may be exposed to higher selection pressures as these loci are often located in regulatory regions. Studies have shown that starting at a certain repeat number, SSRs can acquire mutation rates greater than those of non-repetitive loci and/or loci with just two repeats [34]. Therefore the observed mutation rate in SSR regions may overestimate the overall mutation rate in the bacterial genome.
Line 498: Consider discussing the possibility that there are selective pressures at SSR loci, as this has been suggested extensively in the literature and SSRs are often located in regulatory regions.
AU: see change in the text with the above remark.
Lines 500–502: This seems vague. Can the authors further explain how a strain-specific mathematical model or control program would be modified given the long latency?
AU: Good point, this sentence was removed from the discussion.
Lines 528–530: Yes please! This would be a very strong dataset to explore the evolution and epidemiology of MAP over time using whole genome sequencing.
AU: This is indeed what we are currently working on. Further publications to follow!